# Trust-Aware Fog-Based IoT Environments: Artificial Reasoning Approach

**Mustafa Ghaleb** [1] and **Farag Azzedin** [1,2,*]

1 Interdisciplinary Research Center for Intelligent Secure Systems, King Fahd University of Petroleum and Minerals, Dhahran 31261, Saudi Arabia

2 Information and Computer Science Department, King Fahd University of Petroleum and Minerals, Dhahran 31261, Saudi Arabia

* Correspondence: fazzedin@kfupm.edu.sa

**Abstract:** Establishing service-driven IoT systems that are reliable, efficient, and stable requires building trusted IoT environments to reduce catastrophic and unforeseen damages. Hence, building trusted IoT environments is of great importance. However, we cannot assume that every node in wide-area network is aware of every other node, nor can we assume that all nodes are trustworthy and honest. As a result, prior to any collaboration, we need to develop a trust model that can evolve and establish trust relationships between nodes. Our proposed trust model uses subjective logic as a default artificial reasoning over uncertain propositions to collect recommendations from other nodes in the IoT environment. It also manages and maintains existing trust relationships established during direct communications. Furthermore, it resists dishonest nodes that provide inaccurate ratings for malicious reasons. Unlike existing trust models, our trust model is scalable as it leverages a Fog-based hierarchy architecture which allows IoT nodes to report/request the trust values of other nodes. We conducted extensive performance studies, and confirm the efficiency of our proposed trust model. The results show that at an early stage of the simulation time (i.e., within the first 2% of the number of transactions), our trust model accurately captures and anticipates the behavior of nodes. Results further demonstrate that our proposed trust model isolates untrustworthy behavior within the same FCD and prevents untrustworthy nodes from degrading trustworthy nodes' reputations.

**Keywords:** artificial reasoning; Fog computing; Internet of Things; trust modelling

## 1. Introduction

The Internet of Things (IoT) system is a new ever-growing communication technology in which various nodes such as sensors, smartphones, and devices communicate with one another through the internet [1–4]. IoT environments' continuous growth and advancement have led to a higher level of availability, scalability, accessibility, and interoperability [1,5]. However, these systems are vulnerable to cyberattacks, and hence the number of attacks is exponentially increasing due to various IoT devices and protocols, various attack surfaces, and a lack of security and trust standardizations and requirements [6,7].

The rapidly growing concerns for security and trustworthiness of IoT-based systems/services have paved the way for researchers and practitioners to build trusted IoT environments [8–12]. Cyber intelligence research groups [13–15] have likewise pointed towards artificial intelligence as having a vital role in improving trust modeling and consequently focusing on IoT cybersecurity, including artificial intelligence schemes, usually in terms of modelling unusual behavior. When a separate entity holds each opinion, combining several opinions into a single opinion is a common task for artificial reasoning systems. Subjective logic, a framework for artificial reasoning [16,17], is applicable when the situation to be analyzed is characterized by considerable epistemic vagueness resulting from incomplete knowledge.

One of the prerequisites to providing appropriate IoT service levels is the need to consider the level of certainty and knowledge related to received information and the level of trust we have in the information source. Maintaining and updating a trust parameter specifically for the information source is one technique to ensure that [18,19]. Increasing the amount of total work completed through the collection of resources is one of the goals of resource sharing [20,21]. Sharing resources in a hostile environment necessitates strict security precautions to safeguard customers and the resources [22]. The costs associated with security provisioning may outweigh the performance benefits that resource sharing is intended to achieve. The sharing relationships can be limited to mutually trusting entities if the predicted trust levels between the various entities are known; as a result, the overhead associated with security provisioning can be decreased. IoT systems provide customers the right to access resources or services that would not otherwise be accessible to them [23,24]. However, the risk involved in the concept of sharing resources or services makes the idea of having a virtual network structure unattractive [25]. Such clients choose to utilize their own closed box due to the vitality and sensitivity of the data or information. This kind of resource management is not only expensive but also ineffective. Clients and resources perceive such an environment as providing both possibilities and risks [24,26]. Security threats, privacy concerns, and susceptibility to harm from malicious attacks are factors that impact people's trust in entities [3,24,26]. Malicious nodes are not trustworthy as defined in [3]. A node is trustworthy if there is a firm belief in the competence of the device to act as expected such that this firm belief is not a fixed value associated with the device but rather it is subject to the device's behavior and applies only within a specific context at a given time [3]. Trustworthiness-establishing techniques must be developed in this world of risk and uncertainty. In order to help entities determine how much trust to place in others, such techniques should be used. As a result, it is crucial to create trustworthy IoT settings [4,9,11]. Building trustworthy IoT environments also reduces irreparable and unforeseen losses, developing dependable, effective, and steady service-driven IoT systems. We cannot assume that every node in widely distributed networks is aware of every other node or that every node is reliable.

In this article, we use subjective logic to model trust relationships between IoT entities. A standard task in artificial reasoning systems is to fuse opinions into a single opinion. Artificial reasoning is also applicable when the situation to be analyzed is characterized by considerable epistemic uncertainty due to incomplete knowledge. Subjective logic is particularly appealing for applications in artificial intelligence as well as other fields such as reliability analysis and information security due to its conciseness and simplicity [16,17]. Hence, the subjective logic method can be utilized for trust reasoning [17].

The rest of the article is organized as follows. Section 2 outlines the current IoT trust models. For completeness and clarity purposes, we discuss the components of trust model architecture in Section 3. Section 4 presents the subjective logic trust model for IoT systems. The performance evaluation experiments are discussed in Sections 5 and 6. Finally, Section 8 concludes the article and envisions future directions.

## 2. Related Work

This section discusses the research progress of building trust models in Fog computing environments. Recently, few studies have considered building trust models in Fog computing environments. These studies use different approaches to manage trust functions (e.g., trust computation and trust information records). These approaches can be broadly classified into three categories: centralized, distributed, or hybrid. In the centralized approach, trust functions are managed by a Fog broker [27], Fog assist [28], or cloud [29]. In the distributed approach [10,30], however, the participated Fog nodes (either Fog or thing node) are involved in handling the trust functions. In a sense, the participated nodes calculate the trust level of other nodes of interest and maintain the trust table locally or globally by sending it periodically or on-demand basis. In the Hybrid approach, the trust functions are managed and deployed in the things layer, the Fog, or the cloud layer [31,32].



### 2.1. Fog-Based Centralized Trust Models

Several studies use the centralized approach such as [27–29,33]. Rahman et al. [27] use a central entity called a Fog broker to maintain and compute the trust level of the participating nodes in a Fog computing environment. A fuzzy logic-based method evaluates the trust level of a particular node. It is calculated based on QoS, availability, security considerations, user feedback, and cost. The availability parameter of the node is used in the evaluation process to provide end-users with an available trustworthy Fog that satisfies their requirements. Junejo et al. [28] propose a trust management system for Fog-enabled cyber-physical systems. The direct trust level of Fog nodes is assessed using QoS and network-related parameters. In this study, trust computation is formulated as a statistical regression problem. Subsequently, it is solved using random forest regression. They conclude that the credibility and multi-factor trust evaluations enable precise and accurate trust computation and guarantee a trustworthy Fog-enabled CPS system.

Zineddine et al. [33] developed a trust model using fuzzy neural networks and the weighted weakest link (WWL) to evaluate the trustworthiness of Fog nodes. After passively classifying all Fog nodes using a neuro-fuzzy classifier and the WWL algorithms, the results are sent to a central entity called the Fog trust server. The used model enables the end-user to select the most trusted node upon requesting a service of interest. The information of all nodes needs to be collected and processed offline. Hosseinpour et al. [34] propose a framework based on role-based access control and trust model for the Fog computing environment. In this study, a new Fog node is assigned a trivial task, and the model computes a node's trustworthiness based on the allocated task's satisfaction level. Once the trusted Fog nodes are discovered, they are given more access privileges and roles. The overall trust level of a Fog node is computed based on the composition of different trust attributes such as reliability, availability, turnaround efficiency, and data integrity.

Guo et al. [29] propose a three-tier trust model, cloud-cloudlet-device, based on social relationships among IoT devices owners. The proposed model employs a cloudlet as an intermediary layer for maintaining trust data. The calculation of the trustworthiness of all involved IoT devices is performed by the cloud in the region of cloudlets. Al Muhtadi et al. [35] propose a trust model to enhance IoT security based on subjective logic. The proposed system computes the trust level for each Fog node and keeps the trust level associated with Fog Id in the local list. Based on the aggregated trust values, the system determines if a Fog node is a legitimate node or a rogue node. Zahra and Chishti [9] develop a Generic and Lightweight Security mechanism utilizing artificial reasoning called Fuzzy logic and Fog computing technology called GLSFIoT. It integrates the benefits of fog computing and fuzzy logic into designing a minimal security solution for IoT. The cloud layer nodes employ anomaly-based and fuzzy logic-based trust management and detection techniques to spot malicious activity in the fog layer in the IoT environment and pinpoint the most reliable fog nodes. Additionally, they stated that combining many edge nodes with reliable fog nodes is crucial before implementing security measures to minimize load and improve system performance. The trustworthy fog nodes carry out processes for flagging and detecting unknown attacks.

### 2.2. Fog-Based Distributed Trust Models

Several studies use the distributed approach such as [10,30,36]. Al-Khafajiy et al. [30] design a Fog COMputIng Trust manageMENT (COMITMENT) system. They consider direct and indirect trust in calculating the overall trust level of the participating nodes. In the direct trust, QoS (e.g., low latency) and quality of protection (QoP) (e.g., data protection) metrics are used to assess the trust level of selected Fog nodes by adopting the Bayesian network method. In the indirect trust, recommendations from neighboring Fog nodes are used to assess the trustworthiness of participated Fog nodes by adopting the distributed Collaborating Filtering method. In this system, two types of recommenders, namely trusted Fog nodes and community Fog nodes, are considered to evaluate the relationships between the trustor Fog node and the recommenders. If the recommender is a trusted Fog node

with a satisfactory experience score, the recommendation will be considered; otherwise, the recommendation will be ignored. Whereas if the recommender is a community Fog node, it will be considered if it has similar standards (e.g., same QoS and QoP experience) with the trustor.

Alemneh et al. [10] develop a two-way TMS based on subjective logic. In this study, three QoS parameters were utilized by the service provider (Fog server) to evaluate the truthfulness of the service requester (Fog client), namely friendship, honesty, and ownership. At the same time, the Fog client evaluates the trust level of the Fog server in delivering stable and secure services by utilizing three social relationship metrics: latency, ownership, and packet delivery ratio.

Kouicem et al. [36] propose a hierarchical trust management architecture based on blockchain with mobility support in the Internet of Everything (IoE)-based systems. The trust values of IoT devices are stored in the blockchain maintained by distributed Fog nodes. The trust model considers the honesty of IoT devices to report recommendations to their managing Fog nodes about other service providers. So, they only focused on recommendations and ratings.

Wang et al. [37] propose a TMS for mobile Fog computing services. In this model, three parameters are used to evaluate the direct trust of sensor nodes: residual energy, node communication interaction, and packet loss rate. The overall trust score considers recommendations with direct trust.

Ogundoyin and Kamil [11] propose a bi-directional trust management system to fulfill both secure offloading and fog-to-fog collaboration. The model allows a service requester to calculate the trustworthiness of the service providers and vice versa before any transaction. The proposed model uses QoS, level of security, and social relationships to compute the trust level of a node by fusing these trust parameters with past reputation and recommendations from other neighboring nodes.

Gao et al. [38] propose a lightweight multidimensional trust evaluation mechanism in the service-oriented IoT edge environment. The overall trust in the proposed model is developed to assess the IoT edge devices, which are divided into three parts: direct trust, capability trust, and indirect trust from feedback. Furthermore, a double-filtering mechanism based on multi-source feedback is developed to ensure the correctness and reliability of indirect trust. Finally, the trust evaluation process is completed by IoT edge devices and the edge server without the involvement of the cloud server.

Elmisery et al. [39] offer a Fog-based middleware in which trust agents compute the approximated interpersonal trust between a Fog node and the Cloud. The entropy definition is used to perform the trust computation in a decentralized manner. In order to improve and maintain trust, the Fog architecture uses a service layer on top of the Fog.

Hussain and Huang [40] present a Trust and Reputation-based Model called TRFIoT for Fog-based IoT to evaluate the trustworthiness of Fog nodes. They employed a multi-source trust assessment that considered the reputation of participating nodes. Authors also use periodic trust feedback to make the system trustworthy and reliable. The local content is assessed in the Fog layer to inspect malicious nodes, while global content is assessed in the cloud layer.

Singh et al. [41] propose a Robust Trust Management scheme for Edge of Things in smart cities based on Bayesian learning and collaboration filtering. The evaluation feedback from users and IoT devices is aggregated in the edge servers, and the final result is sent to the cloud server.

Compared to existing trust mechanisms, Yuan et al. [42] present a reliable and lightweight trust model specified for IoT edge devices by employing objective information entropy theory. Their method for computing the global trust is based on fusing multi-source feedback information. The broker and device layers complete the trust computation in this case. The authors chose the lightweight trust evaluation method because it is well-suited for large-scale IoT environments. Jain and Kumar [12] incorporated blockchain into an IoT context with fog technology. They suggest a trusted task offloading system that ensures the

quality of service for IoT consumers while operating in a decentralized manner. They used subjective logic to calculate the trust value for the fog nodes.

### 2.3. Fog-Based Hybrid Trust Models

Wang et al. [31] propose a Fog-based hierarchical trust model to evaluate trustworthiness in sensor-cloud systems. In this model, the behavior trust is formed among nodes in the wireless sensor layer, and the data trust of nodes is formed in the Fog layer. The Fog layer plays a significant role and acts as a trust buffer zone between the things layer and the cloud layer. This model is based on direct interaction and recommendations from other nodes. It constructs the trust level in the Fog layer as follows. First, the Fog layer obtains the trust state of the IoT things periodically. Then, the Fog layer performs data analysis to reveal hidden data attacks and ensure the credibility of edge nodes. Finally, the Fog layer establishes trust relationships between cloud service providers and sensor service providers through collecting pieces of evidence. Zhang et al. [32] propose a three-layer Fog-based detection system that utilizes a trust evaluation method to detect and solve hidden data attacks. The first is the direct trust layer in Wireless Sensor Networks (WSNs). The second layer is the preliminary decision layer among underlying Fog nodes, and the third layer is the data analysis layer in the Fog platform. Their approach for evaluating the trustworthiness of nodes is based on direct trust and recommendation. The considered trust metrics are packet success rate, forwarding delay, and routing failure rate.

Jabeen et al. [43] present a hybrid trust management system that uses centralized and distributed techniques to assess the trustworthiness of nodes. The nodes execute in-network processing and convey only the fine-grained values to the fog node for assessment to satisfy the scalability requirement in the centralized method. The fog nodes forward their calculated reputation values of IoT devices to the cloud for global reputation computation. The proposed model meets the adaptivity, survivability, and scalability requirements of the trust management system for the IoT environment.

A trust management method IoT edge computing system in smart cities is presented by Wang et al. [44]. The evolutionary game theory improves the validity and stability of the trust management system. The validity and stability of the trust management system are proved using the Lyapunov theory. The suggested approach enhances IoT device collaboration in edge computing. Their approach primarily examines the trust relationships of end-users. The approach is then extended to include end users and Edge service providers. They maintain a blocklist and allowlist methods as part of their method. By the end nodes, the reputable nodes will be added to the allowlist, while the suspicious nodes will be added to the blocklist. During path selection, the network tries to entail the nodes in the allowlist.

### 2.4. Summary

To conclude, Table 1 gives a summary of the current Fog-based trust model approaches that address the concept of trust. Models can be categorized in terms of (a) the hosting environment or (b) the trust characteristics when calculating trust. Direct trust and/or reputation are the two factors that trust models use to calculate trust. If reputation is utilized to calculate trust level, filtering mechanisms are used to weed out recommenders or adjust/shift recommendations. This filtering process can be based on attributes such as accuracy and honesty. A trust model is utilized to specify, annotate and build trust relationships between IoT nodes for the purpose of intelligent reasoning. There are different types of intelligent reasoning methods and Table 2 classifies Fog-based trust models according to these methods, while Table 3 identifies the tool used in simulating and evaluating the proposed Fog-based trust model.

**Table 1.** Fog-based Trust Models.

| Ref., Year | Hosting Environment | | | | | Trust Characteristics | | | | | |
| --- | --- | --- | --- | --- | --- | --- | --- | --- | --- | --- | --- |
| | IoT Layer | | | Architecture | | Trust Update | | Trust Components | | Trust Filtering Mechanisms | |
| | Things | Fog | Cloud | Cen. | Dis. | Time | Event | Direct | Reputation | Recommenders | Recommendations |
| [31], 2018 | ✓ | ✓ | - | ✓ | ✓ | ✓ | - | ✓ | ✓ | ✓ | ✓ |
| [37], 2019 | - | ✓ | - | - | ✓ | - | ✓ | ✓ | ✓ | ✓ | - |
| [28], 2019 | ✓ | ✓ | - | ✓ | - | - | ✓ | ✓ | - | - | - |
| [30], 2020 | - | ✓ | - | - | ✓ | ✓ | - | ✓ | ✓ | ✓ | ✓ |
| [10], 2020 | - | ✓ | - | - | ✓ | - | ✓ | ✓ | ✓ | ✓ | ✓ |
| [27], 2020 | - | ✓ | - | ✓ | - | ✓ | - | ✓ | ✓ | - | - |
| [33], 2020 | - | ✓ | - | ✓ | - | - | - | ✓ | ✓ | ✓ | - |
| [34], 2017 | - | ✓ | - | ✓ | - | - | - | ✓ | - | - | - |
| [32], 2018 | ✓ | ✓ | - | ✓ | ✓ | - | - | ✓ | - | - | - |
| [36], 2018 | ✓ | ✓ | - | - | ✓ | - | - | - | ✓ | ✓ | - |
| [29], 2017 | - | - | ✓ | ✓ | - | - | ✓ | ✓ | ✓ | ✓ | - |
| [35], 2021 | - | ✓ | - | ✓ | - | - | ✓ | - | ✓ | - | ✓ |
| [11], 2021 | ✓ | ✓ | - | - | ✓ | - | ✓ | ✓ | ✓ | ✓ | - |
| [39], 2017 | - | ✓ | ✓ | - | ✓ | - | ✓ | - | ✓ | - | - |
| [40], 2018 | - | ✓ | - | - | ✓ | ✓ | - | ✓ | ✓ | - | - |
| [43], 2021 | ✓ | ✓ | - | ✓ | ✓ | - | ✓ | ✓ | ✓ | ✓ | - |
| [41], 2021 | - | ✓ | - | - | ✓ | ✓ | - | ✓ | ✓ | ✓ | - |
| [42], 2018 | ✓ | ✓ | - | - | ✓ | ✓ | - | ✓ | ✓ | - | - |
| [44], 2020 | ✓ | ✓ | - | ✓ | ✓ | ✓ | - | ✓ | ✓ | - | ✓ |
| [12], 2022 | - | ✓ | ✓ | - | ✓ | - | ✓ | ✓ | ✓ | - | ✓ |
| [9], 2022 | - | - | ✓ | ✓ | - | - | ✓ | ✓ | ✓ | - | - |
| Ours, 2022 | ✓ | ✓ | - | ✓ | ✓ | - | ✓ | ✓ | ✓ | - | ✓ |

Ref. = Reference, Cen. = Centralized, Dis. = Distributed.

**Table 2.** Fog-based Trust Models: Reasoning Methods.

| | Reasoning Method | | | | | | |
| --- | --- | --- | --- | --- | --- | --- | --- |
| | Bayesian Systems | Fuzzy Logic | Subjective Logic | Entropy Theory | Game Theory | Beta Distribution | Not Applicable |
| Ref. | [30] | [9,11,27,33] | [10,12,35] | [39,42] | [44] | [38] | [28,29,31,32,34,36,37,40,43] |

**Table 3.** Fog-based Trust Models: Simulation Tools.

| | Simulation Tool | | | | | | |
| --- | --- | --- | --- | --- | --- | --- | --- |
| | Matlab | iFogSim | NS3 | SystemC | NetLogo | Cooja | Not Mentioned |
| Ref. | [11,27,30–33,37,38,40] | [28,35] | [11,29,43] | [34] | [38,42] | [9] | [10,12,36,39,41,44] |

## 3. Trust Model Architecture Components

### 3.1. Trust Representation and Usage

The fundamental trust model concepts have been identified and published in our previous articles [3,25]. These fundamental trust model concepts are: (a) behavior trust, (b) reputation, (c) honesty and (d) accuracy. The trust model elements are organized to function and evolve trust in a fully distributed manner, and no entity is omniscient. Rather, each entity $x$ has its opinion of how trustworthy other entities are and stores this knowledge in its *direct trust table* ($DTT_x$) that contains trust levels for nodes with which $x$ had prior direct experience. For example, $x$ trusts node $y$ to act as expected within context $c$ at time $t$.

Similarly, each node cooperates with other nodes by sharing information in the form of recommendations. In the trust model, node $x$ maintains a *recommender trust table* ($RTT_x$) that associates trust levels for node $y$ based on prior direct experience for node $y$ with other nodes. For example, $T_{ry}^c$ means that $x$ associates a trust level for $y$ based on recommendation received from recommender $r$ based on prior direct experience for $y$ with $n$. This recommendation is for context $c$.

### 3.2. Trust Model Mapping

Since IoT systems can have a large number of nodes, it is not desirable to consider each IoT node as an entity since this will impact the proposed trust scalability. As such and as shown in Figure 1, the IoT system is aggregated into Fog Computing Domains (FCDs).

During this aggregation process, each FCD elects one coordinator. The coordinator represents the entire FCD in the trust modeling process, and it is held responsible for managing its FCD members to maintain a high reputation for its FCD within the IoT environment.

For example, assume that an FCD with six member nodes and one of the members is the coordinator. We will refer to the Fog coordinator as FC. The FC has the complete trust levels of the members within its FCD. Suppose a new node needs to join the FCD. The FC will request it to provide references from prior associations with other FCDs. The objective of the FC is to show its resources as highly trustworthy to other FCs because the value of highly trustworthy resources is higher than the less trustworthy resources. Therefore, the FC does not want to overestimate or underestimate the trust level of a member node because the FC and hence its FCD's reputation will suffer.

Figure 1 shows that each FCD is a collection of member nodes (IoT things) which are a mixture of clients and resources. These members collectively contribute to the overall trustworthiness of their FCD. In contrast to the reputation of other FCDs, which is maintained in $RTT_{FCD}$, the trust records that an FCD believes in other FCDs with whom it has had direct interactions are maintained in $DTT_{FCD}$. These two data structures, associated with each FCD, are maintained by the FC. The member nodes are where the real interaction between two FCDs takes place. The FC also maintains *internal trust table* $ITT_{FCD}$ that contains the trust levels for the members of FCD. By obtaining feedback from FCDs that utilized the service provided by its member node, the FC modifies the member node's trust level in the $ITT_{FCD}$ each time it takes part in a transaction.

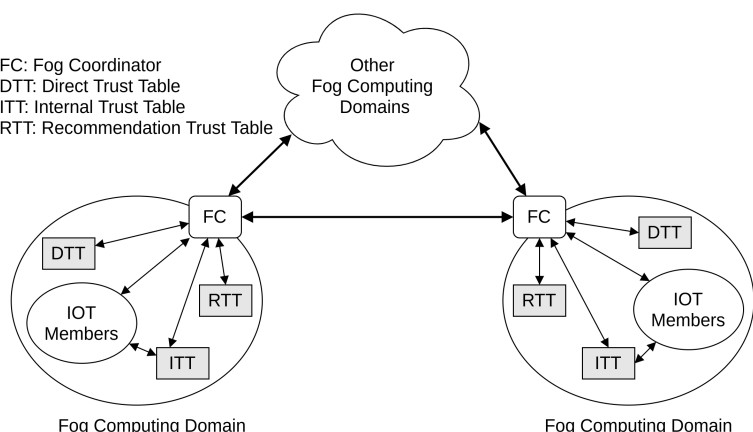

**Figure 1.** Trust model architecture components.

Figure 2 illustrates the operations of the trust model. Assume a source (i.e., FC) wants to determine the trust level of a target (i.e., FC). The source determines the target's trust level by combining its direct trust (i.e., its own experience) with the target as well as the target's reputation. From its DTT, the source obtains its direct trust in the target. The target reputation is obtained by the source contacting its recommenders in RTT. After the recommendations are received, the source can compute the target reputation.

After that, and as shown in Figure 2, the source computes the target trust level by combining its direct trust level in the target and the target reputation. Based on the computed target's trust level, the source can decide to go ahead or reject the transaction with the target. After the transaction is finished, the followings are performed: (a) the source can update its RTT by evaluating its recommenders; (b) the source can update its DTT; and (c) the target can update its ITT to reflect the trustworthiness of its member node.

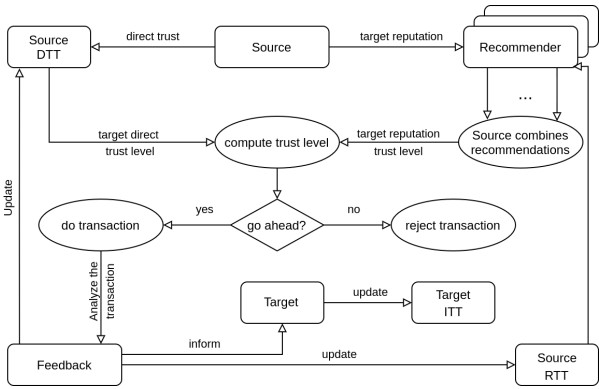

**Figure 2.** Trust and reputation operational cycle.

## 4. Subjective Logic Trust Model

### 4.1. Trust Attributes

Many behavioral characteristics can be observed to evaluate nodes trust level during the nodes interaction process. In the following, we discuss these trust attributes and provide how we calculate them.

Availability: As stated in [34,45,46], it is a measure to ensure that node $j$ is operational and accessible to legitimate nodes whenever needed in a reasonable time to achieve the expectations. Node $j$ might be unavailable if (1) it is busy processing other requests or (2) it refuses to respond to requested task. Availability $A_{i,j}^t$ denotes the availability of $j$, as observed by $i$ at time $t$. $A_{i,j}^t$ is calculated as follows:

$$A_{i,j}^t = \frac{\text{Accepted tasks}}{\text{Total tasks}} \tag{1}$$

Reliability: As stated in [34,45,46], it is a measure to ensure that node consistently operates as per its specifications in a defined time. Reliability $R_{i,j}^t$ represents $j$'s success rate in the completion of the tasks that it has accepted from $i$. $R_{i,j}^t$ is computed based on the following equation:

$$R_{i,j}^t = \frac{\text{Successful performed tasks}}{\text{Accepted tasks}} \tag{2}$$

Turnaround Efficiency: As stated in [34,46], it is a quality measure that ensures $j$ performs a requested task within the time that it promises to $i$. Irregularities and high latency in response time predict potential intrusions in the system. Turnaround is a time frame that begins when $i$ sends a processing task to $j$ until $j$ accomplishes the task successfully and sends it back to $i$. Turnaround Efficiency $E_{i,j}^t$ of $j$ at time $t$ is computed based on the following equation:

$$E_{i,j}^t = \frac{\text{Promised Turnaround time}}{\text{Actual Turnaround time}} \tag{3}$$

If the promised turnaround time is greater than the actual turnaround time, the turnaround efficiency is 1.

Ownership: Every node in the Thing/Fog has an owner. The inclusion of this measure is based on the belief that devices belonging to the same owner have mutual trust. As a result, the value of the trust attribute $Oi, j$ is set to 1 if a node comes across another node that the same owner owns; otherwise, it becomes 0.

### 4.2. Subjective Logic

Subjective logic is emerging as a reasoning method to express trust relationships as a subjective opinion with a degree of uncertainty. Subjective logic, a type of belief theory, is based on the idea that trust is subjective and that everyone experiences it dif-

ferently [10,17,35]. Subjective logic is suggested to be the most appropriate for modeling trust in the Fog computing environment [10,47] as each node computes another node's trust value it encounters subjectively. It is impractical for nodes to include all relevant trust metrics when determining node's trust rating. This means that trust is calculated based on a lack of proof, and that each node calculates its subjective trust levels for each node it meets [10,35]. Opinions are used as input and output variables in the subjective logic method. An opinion $\omega_x$ [48] is defined as:

$$\omega_x = (b_x, d_x, u_x, a_x) \tag{4}$$

where $b_x$ denotes the degree of belief that $x$ is trustworthy, $d_x$ denotes disbelief, $u_x$ denotes uncertainty of trust relation, and base rate $a_x$, atomicity, indicates the prior probability of $x$ in the absence of evidences. The summation of variables $b_x$, $d_x$ and $u_x$ equals 1.

The node computes the aforementioned subjective trust tuple values using the acquired observations and reported experiences from other nodes, as shown in the equations below (5)–(7).

$$b_x = \frac{pos}{pos + neg + k} \tag{5}$$

$$d_x = \frac{neg}{pos + neg + k} \tag{6}$$

$$u_x = \frac{k}{pos + neg + k} \tag{7}$$

where the *pos* and *neg* denote positive and negative acquired observations, respectively, and $k = 1$.

The discount operator $\otimes$ is used to weigh the node's recommendations with the past opinion about the recommender. For example, the discount operator is used when node $i$ intends to compute the reputation values for node $j$ based on a recommendation from the intermediate node $k$ as shown in Figure 3a. In this situation, high trust values are a direct effect of the weight of trusted recommenders and vice versa. Two opinions are averaged using the consensus operator $\oplus$, which is shown in Figure 3b. For more details about $\otimes$ and $\oplus$ refer to [4].

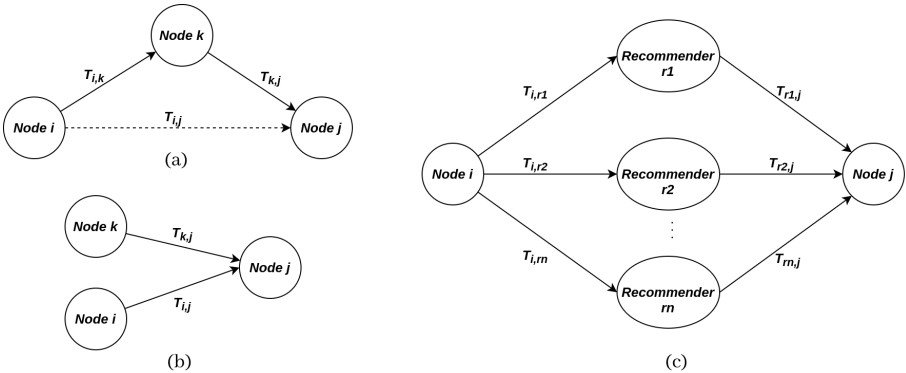

**Figure 3.** (**a**) Discounting operator, (**b**) Consensus operator, (**c**) Computation of reputation.

Reputation can be acquired by carrying out both operators, $\otimes$ and $\oplus$, on the collected recommendations as a subjective trust. This scenario has been illustrated in Figure 3c. Suppose node $i$ has a list of recommenders with the following trust levels at time $t$ as $T_{i,r1}^t, T_{i,r2}^t, \ldots, T_{i,rn}^t$ and these recommenders have trust levels about $j$ at the time $t$ as $T_{r1,j}^t, T_{r2,j}^t, \ldots, T_{rn,j}^t$, then once the discounting and consensus operators have been applied, the reputation of $j$ is calculated as:

$$R_{i,j}^t = (T_{i,r1}^t \otimes T_{r1,j}^t) \oplus (T_{i,r2}^t \otimes T_{r2,j}^t) \quad \oplus \ldots \oplus (T_{i,rn}^t \otimes T_{rn,j}^t) \tag{8}$$

In order to calculate the direct trust of the $j$, the aforementioned trust attributes are utilized. The direct trust of $j$ as evaluated by $i$ at time $t$ is computed utilizing the following Formula (9):

$$T_{i,j}^t = \beta A_{i,j}^t + \gamma R_{i,j}^t + \delta E_{i,j}^t + \eta O_{i,j} \tag{9}$$

where $0 \leqslant (\beta, \gamma, \delta, \eta) \leqslant 1$ and $(\beta + \gamma + \delta + \eta) = 1$. The weighting factors of availability and reliability should be set to high values to encourage nodes to provide a reliable service whenever needed. The detailed steps of the trust computation and updating trust-related tables are described in Algorithm 1. The trust level $\Psi_i^j$ is in range of [0 , 1], where 1 indicates complete trust and 0 is complete distrust.

---

**Algorithm 1** Computing and Updating Trust-related Tables.

---

**Require:** Node $i$ as source, Node $j$ as target, $S_i$ as service, $\lambda$, Threshold
**Ensure:** Trust Level of $j$
 1: Retrieve list of $j$s that provide $S_i$
 2: **for each** $j$ **do**
 3:     **for each** $r \in RTT_i$ **do**
 4:         Acquiring recommendations for $j$
 5:         Apply discounting operator
 6:         Apply consensus operator
 7:     Apply discounting and consensus operators using Equation (8)
 8:     $\Psi_i^j = \lambda \times T_{i,j}^t + (1 - \lambda) \times R_{i,j}^t$ ▷ calculate trust level $\Psi_i^j$ by combining the direct trust and reputation
 9:     **if** $\Psi_i^j \geqslant Threshold$ **then**
10:         Do transaction with $j$
11:         Go to line 12
12: Compute direct trust $T_{i,j}^{t+1}$ using Equation (9)
13: **if** $T_{i,j}^{t+1} \geqslant Threshold$ **then**
14:     $pos_{i,j} = 1$
15: **else**
16:     $neg_{i,j} = 1$
17: **if** $j \notin FCD_i$ **then**
18:     Send feedback to $FC_j$
19:     Update $ITT$ by $FC_j$
20: $T_{i,j}^t = \lambda \times T_{i,j}^t + (1 - \lambda) \times T_{i,j}^{t+1}$                                             ▷ update $DTT_i$
21: **for each** $r \in RTT_i$ **do**                                                                            ▷ update $RTT_i$
22:     Update trust level for $r$

---

Algorithm 1 starts with the member node that wants a specific service by contacting its coordinator and asking for service providers that provide the required service. The coordinator utilizing the inter-ring, sends a lookup query and gets a list of all service providers. The coordinator selects a trustworthy provider by consulting its DTT and the recommendations from its recommenders. Next, the coordinator receives the recommendations in the form of opinions. These recommendations are weighted and combined to get the final reputation of the coordinator of the service provider by applying discounting and consensus operators together. Then, the coordinator computes the trust level $\Psi_i^j$ based on the reputation and the direct trust with the coordinator of the service provider. Based on the aggregated results, the coordinator decides to go with the transaction or not if $\Psi_i^j \geqslant Threshold$. After doing the transaction, the $T_{i,j}^{t+1}$ of the service provider is computed using Equation (9). If the $T_{i,j}^{t+1} \geqslant Threshold$, it assigns positive feedback; otherwise, it assigns negative feedback. Then, the feedback is sent to the Fog coordinator of the service provider to update its internal trust table. Finally, the coordinator of $i$ updates its $DTT_i$ and $RTT_i$.

### 4.3. Trust Model Elements

Tables 4–6 represent the tables maintained by each *FC*, namely $DTT_{FC}$, $RTT_{FC}$ and $ITT_{FC}$. These tables utilize subjective logic parameters to compute and update trust levels. Table 4 is maintained by Fog Coordinator $x$ ($FC_x$) for Fog Coordinator $y$ ($FC_y$). Similarly, Tables 5 and 6 are maintained by $FC_x$.

**Table 4.** Direct Trust Table maintained by $FC_x$ for $FC_y$.

| Context | Number of Transactions | | Subjective Logic Parameters | | | | Trust Level | Time Stamp |
|---|---|---|---|---|---|---|---|---|
| | + | − | Belief | Disbelief | Uncertainty | Atomicity | | |
| $C_1$ | $P^y_{C_1}$ | $N^y_{C_1}$ | $B^y_{C_1}$ | $D^y_{C_1}$ | $U^y_{C_1}$ | $A^y_{C_1}$ | $TL^y_{C_1}$ | $TS^y_{C_1}$ |
| ⋮ | ⋮ | ⋮ | ⋮ | ⋮ | ⋮ | ⋮ | ⋮ | ⋮ |
| $C_i$ | $P^y_{C_i}$ | $N^y_{C_i}$ | $B^y_{C_i}$ | $D^y_{C_i}$ | $U^y_{C_i}$ | $A^y_{C_i}$ | $TL^y_{C_i}$ | $TS^y_{C_i}$ |

**Table 5.** Recommender Trust Table maintained by $FC_x$ for nodes directly interacted with $FC_y$.

| Context | Recommender | Subjective Logic Parameters | | | | Trust Level | Time Stamp |
|---|---|---|---|---|---|---|---|
| | | Belief | Disbelief | Uncertainty | Atomicity | | |
| $C_1$ | $R_1$ | $B^{1y}_{C_1}$ | $D^{1y}_{C_1}$ | $U^{1y}_{C_1}$ | $A^{1y}_{C_1}$ | $TL^{1y}_{C_1}$ | $TS^{1y}_{C_1}$ |
| | ⋮ | ⋮ | ⋮ | ⋮ | ⋮ | ⋮ | ⋮ |
| | $R_n$ | $B^{ny}_{C_1}$ | $D^{ny}_{C_1}$ | $U^{ny}_{C_1}$ | $A^{ny}_{C_1}$ | $TL^{ny}_{C_1}$ | $TS^{ny}_{C_1}$ |
| ⋮ | ⋮ | ⋮ | ⋮ | ⋮ | ⋮ | ⋮ | ⋮ |
| $C_i$ | $R_1$ | $B^{1y}_{C_i}$ | $D^{1y}_{C_i}$ | $U^{1y}_{C_i}$ | $A^{1y}_{C_i}$ | $TL^{1y}_{C_i}$ | $TS^{1y}_{C_i}$ |
| | ⋮ | ⋮ | ⋮ | ⋮ | ⋮ | ⋮ | ⋮ |
| | $R_n$ | $B^{ny}_{C_i}$ | $D^{ny}_{C_i}$ | $U^{ny}_{C_i}$ | $A^{ny}_{C_i}$ | $TL^{ny}_{C_i}$ | $TS^{ny}_{C_i}$ |

**Table 6.** Internal Trust Table maintained by $FC_x$.

| $FCD_x$ Members | Subjective Logic Parameters | | | | Trust Level |
|---|---|---|---|---|---|
| | Belief | Disbelief | Uncertainty | Atomicity | |
| 1 | $B_1$ | $D_1$ | $U_1$ | $A_1$ | $TL_1$ |
| ⋮ | ⋮ | ⋮ | ⋮ | ⋮ | ⋮ |
| $j$ | $B_j$ | $D_j$ | $U_j$ | $A_j$ | $TL_j$ |

## 5. Performance Evaluation

### 5.1. Goals of the Simulation

Cross-ratings are a key component of trust-based systems, and since these systems are built on communities that might contain untrustworthy nodes, it is essential for any trust model to predict the trust level among nodes accurately. Therefore, the first objective is to distinguish between trustworthy and untrustworthy nodes. As a result, a node will be able to conduct transactions with trustworthy nodes and stay away from untrustworthy ones. We investigated this attribute in our trust model and in simulations by evaluating the accuracy of our trust model in predicting the trust between nodes. In addition, we vary the weight of direct trust versus reputation to scrutinize the agility of the trust model in detecting untrustworthy nodes. Furthermore, we vary the number of dishonest nodes to study the resilience of the trust model to dishonest nodes that give incorrect reviews to mislead predicting the future behavior of nodes for malicious reasons. Hence, the goals of the simulation can be summarized as follows:

- Examine the agility of the trust model in detecting untrustworthy nodes

- Study the effect of dishonest nodes that give incorrect reviews.

### 5.2. Simulation Environment

The trust model is implemented using Contiki OS version 3.0, and the evaluation is performed through the Cooja emulator. For a deployed architecture, we run a set of experiments that follow the same methodology to ensure their valid comparability. The phases of running experiments start by utilizing the Cooja simulator to:

- Select a radio medium model such as distance/constant loss Unit Disk Graph Model (UDGM), Directed Graph Radio Medium (DGRM), or Multi-path Ray-tracer Medium (MRM).
- Select a mote type from the supported motes such as Z1, Wismote, MicaZ, or SKY mote and select the number of motes. In our study, we have used Wismote and SKY mote.
- Select a network topology such as uniformed 2D-Grid or random positioning. More detail about our selected topology comes later.
- Select the transmission range for the populated nodes.

Then, the modules of the trust model are implemented in Cooja and follow Algorithm 1 for constructing the model by computing and updating trust-related tables. After selecting the performance evaluation metrics, Cooja simulation scripts and other tools such as Gnuplot were used to collect, save, analyze, and display simulation results.

Regarding the network topology, we consider the topology shown in Figure 4 as a physical topology for the scenarios under study. The network consists of 4 nodes as FCs and 16 nodes as members of the FCDs. The nodes are deployed in an area of size 150 m × 150 m. The UDGM is used with a transmission range of 50 m and an interference range of 100 m. At the transport layer, UDP is used as a default protocol. The Wisemote platform is used for experimenting with Fog coordinators, while the Sky mote platform is used for experimenting with members of FCDs.

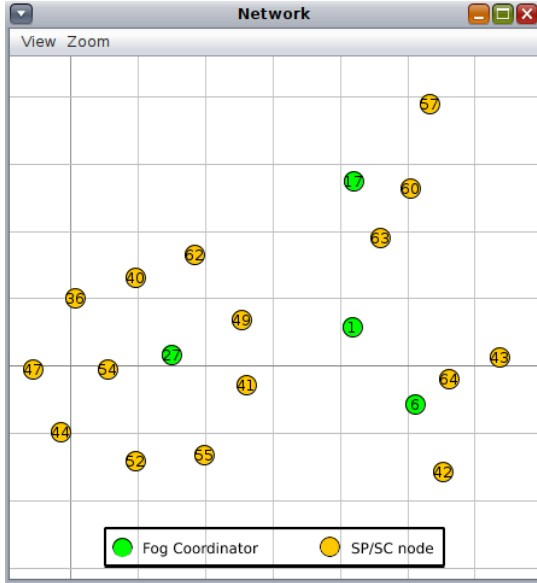

**Figure 4.** Network Topology.

The simulated environment consists of three FCDs. The first FCD has 10 members (i.e., IoT devices) and one FC (i.e., node id 27). There are 3 members in each of the other two FCDs having FCs 6 and 17. Various experiments have been conducted to evaluate the evolution of trust based on different scenarios. Each point in the figures is the result of five simulation runs. Table 7 shows the design parameters used in the simulation.

**Table 7.** Design parameters used in the simulation.

| Symbol | Definition | Design Parameter Values |
|--------|-----------|------------------------|
| $FCD_{num}$ | Number of FCDs | 3 |
| $Nodes_{num}$ | # of members | 16 |
| $B$ | Belief | 0.5 |
| $D$ | Disbelief | 0.5 |
| $U$ | Uncertainty | 0.5 |
| $A$ | Atomicity | 1 |
| $\lambda$ | Direct trust weight | $[0.0, 0.5, 0.7, 0.9]$ |
| $Dis_{num}$ | # of dishonest nodes | $[0\%, 10\%, 30\%, 50\%]$ |
| $Unt_{num}$ | # of untrustworthy nodes | $[0\%, 10\%, 30\%, 50\%]$ |

## 6. Results and Discussion

### 6.1. Agility of the Trust Model

Figure 5 shows the trust evolution when all recommenders are honest. In these experiments, $FC_{17}$ predicts the trust level of $FC_{27}$. During these sets of experiments, we also varied the value of $\lambda$ and the percentages of malicious (i.e., untrustworthy) nodes. It should be noted that GT-Normal means all nodes are trustworthy and GT-Malicious means all node are malicious (i.e., untrustworthy).

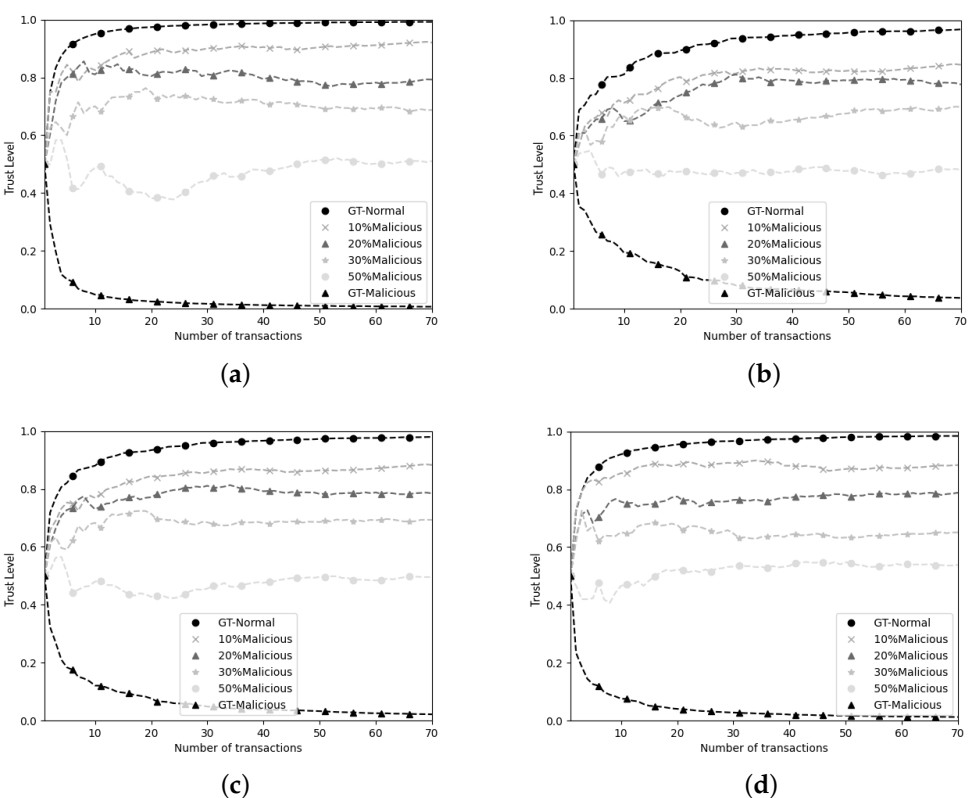

**Figure 5.** For zero dishonest nodes, evolution of trust where $\lambda$ value is: (**a**) 1, (**b**) 0, (**c**) 0.5 and (**d**) 0.7.

Since there are zero dishonest recommenders, it can be observed that combining direct trust and reputation (i.e., when $\lambda \neq 0$) outperforms trust prediction when $\lambda = 0$. Because all recommenders are honest, reputation reinforces direct trust, and therefore combining these two components yields a higher trust level prediction than relying on only one of them. For example, when $\lambda = 0.5$ and 0.7, the trust level converges much faster than when $\lambda = 0$.

It should be noted that this fact holds true for all percentages of malicious (i.e., untrustworthy) nodes. Since we are relying on honest recommenders (i.e., recommenders that

do not lie), reputation reports are not false and therefore the trust level computation is not polluted. Ideally, trust models wants to hinder dishonest recommenders from contributing to trust level calculation. As such, the impact of $\lambda$ is almost negligible on predicting the trust level. This is inline with our expectation since the environment under study consists of honest recommenders.

In the next set of experiments (i.e., Figure 6), we investigate the impact of dishonest recommenders that attempt to contaminate the IoT environment by intentionally providing fake recommendations. We measure such impact while varying the value of $\lambda$. In these experiments, we evaluated the Global (i.e., combining direct trust and reputation), Direct (i.e., only using the direct trust component), and Reputation (i.e., only using the reputation component). This is basically the evaluation of Algorithm 1 line #9. The reason for this is to evaluate the effect of the two components (i.e., direct trust and reputation) on the prediction of the trust level (Global). Fog 6 is configured as a dishonest recommender, and all member nodes of $FCD_{27}$ are malicious (i.e., untrustworthy).

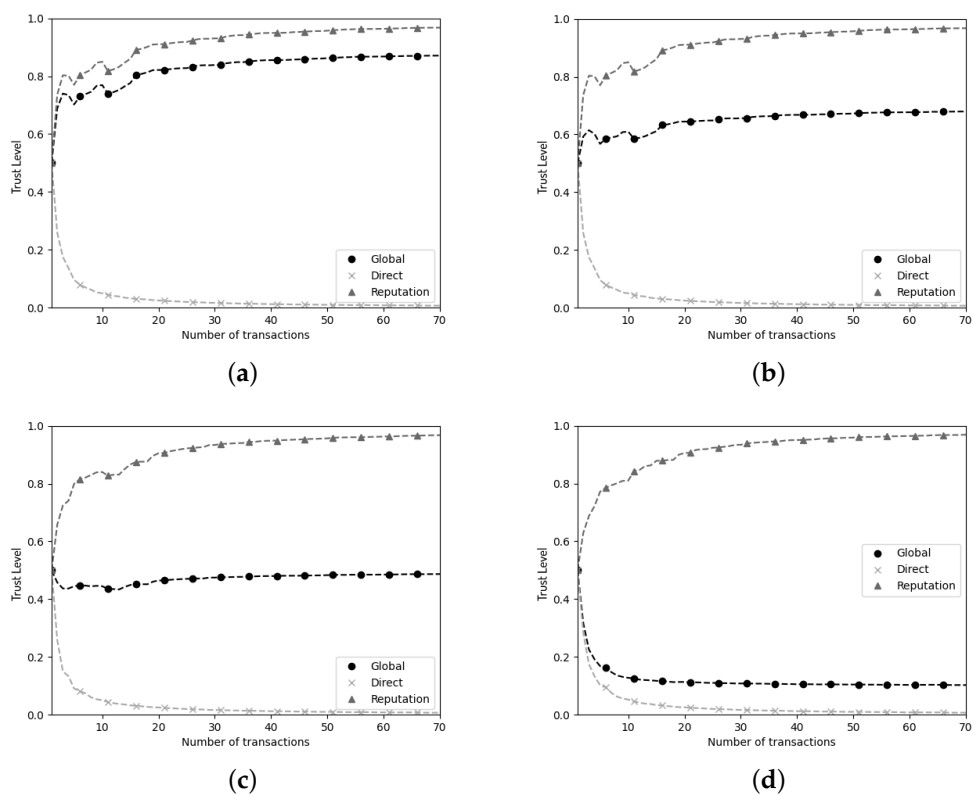

**Figure 6.** Evolution of trust when there are dishonest recommenders where $\lambda$ value is: (**a**) 0.1, (**b**) 0.3, (**c**) 0.5 and (**d**) 0.9.

From the obtained results, increasing $\lambda$ reduces the effect of dishonest recommenders as shown in Figure 6d. On the other hand, when $\lambda = 0.1$ (shown in Figure 6a), the predicted trust level of $FCD_{27}$ is above 0.75 meaning that $FCD_{27}$ is trustworthy while all of $FCD_{27}$ members are untrustworthy. This wrong prediction is caused by giving more weight to the reputation component in calculating $FCD_{27}$ trust level.

Therefore, we can conclude that relying on direct trust converges to an acceptable predicted trust level. Relying on direct trust, however, does not exploit cooperation which is a main goal of IoT systems. On the other hand, a reputation-based model can converge to a high success rate but as the number of dishonest recommenders increases, the trust model becomes sensitive to these dishonest recommendres.

*6.2. Evolution of Internal Trust*

In this section, we have conducted a set of experiments based on one scenario where there are 50% malicious (untrustworthy) member nodes in one FCD to evaluate the effect of the internal trust levels of the member nodes of the FCD. Whenever a member node takes part in a transaction, its FC updates the member node's trust level in its *ITT* utilizing EWMA after discounting the received feedback. The discounting process is based on direct trust with the FC who sent the feedback. We have simulated this scenario in Figure 7.

Figure 7 shows that the network exhibits two types of behaviors. We plot the behavior of trustworthy member nodes of $FCD_{27}$ in Figure 7a. According to the results, the trustworthiness of these trustworthy node members is increasing as the number of transactions increases, and this increase is unaffected by the other 50% unreliable node members in the same FCD (i.e., $FCD27$).

We then plot the behavior of untrustworthy member nodes of $FCD_{27}$ in Figure 7b. The results show that as the number of transactions increase, the trust level of these untrustworthy node members is decreasing and this decrease is not affected by the trustworthy behavior of the other 50% trustworthy node members in the same FCD (i.e., $FCD_{27}$).

This is an important result that shows that the individual behavior of member nodes are captured and isolated their *FC*. Hence, an incentive (i.e., reward/punishment) mechanism can be utilized by the *FC* to further discourage or even prevent untrustworthy behavior in its *FCD*.

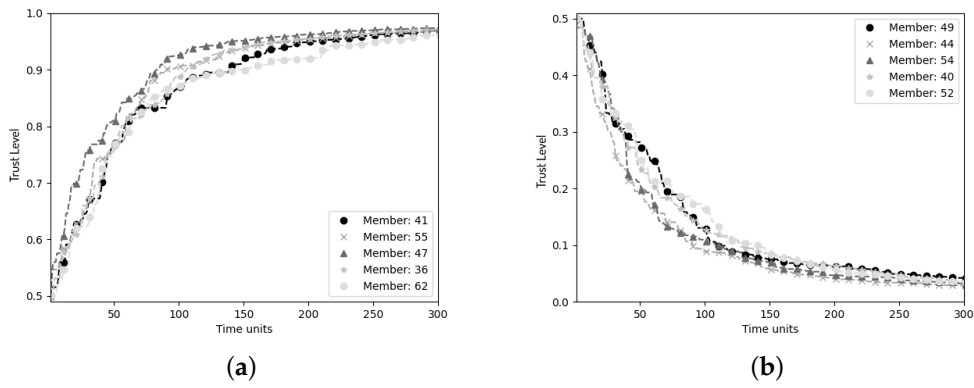

**Figure 7.** Evolution of internal trust of members, (**a**) Trustworthy Members (**b**) Untrustworthy Members.

## 7. Trust Model Realism Furthermore, Limitations

For scalability purposes, the IoT system is aggregated into FCDs. The trust model is designed with the scalability factor in mind. The trust model elements such as DTT, RTT, and ITT are designed to operate and evolve trust in a purely distributed manner. There is no FCD that is omniscient. Rather each FCD: (a) has its own view of how trustworthy other FCDs are and keeps this information in DTT (b) controls the monitoring process of its own transactions using its FC, (c) maintains RTT using its own FC.

The size of RTT is small since it contains information about the FCD recommenders which is a small set of the total number of FCDs. On the other hand, DTT contains one entry for each of the FCDs that this particular FCD directly interacted with. Hence, as the direct interaction with different NCDs increase, the size of DTT increases in a linear fashion.

A potential implementation problem with our trust model is ensuring that the identities of FCDs cannot be created trivially. This is crucial since reputation can get erased if an FCD changes its identity. Hence, an untrustworthy FCD can use this trick to start fresh every time it builds up a bad reputation history. Further, FCDs may expect their recommendation requests to be fulfilled by others. However, when it comes to others asking for recommendations, this particular FCD chooses not to give recommendations by simply returning −1 (i.e., do not know) or basically ignoring the request. That is, FCDs may refuse to give recommendations for various reasons. Further, by isolating dishonest

recommenders and routing the recommendation requests to only honest recommenders, a tedious task can be potentially created by bombarding these honest recommenders with a rather huge volume of recommendation requests. As a result, an honest recommender might be inclined to refuse giving recommendations to avoid the extra work. To remedy this problem, incentives should be provided to encourage and reward these cooperative FCDs. As such, more explicit incentives should be provided by our trust model. For example, cooperative FCDs should be given reduced cost when using resources or should have a higher priority when submitting tasks.

Another limitation of our trust model that we foresee is the potential creation of performance bottlenecks. Because every FCD wants to keep honest recommenders, the majority, if not all, of the recommendation requests will be routed to the honest recommenders. This not only creates a tedious task that can be bothersome for these honest recommenders, but also can create a potential performance bottleneck. Therefore, responses to recommendation requests can experience a longer delay.

## 8. Conclusions and Future Directions

In this article, we proposed a system to model trust relationships in IoT environments. The proposed trust model uses subjective logic since it is particularly appealing for applications in artificial intelligence as well as other fields such as reliability analysis and information security due to its conciseness and simplicity. Hence, the subjective logic method is utilized in our proposed model for trust reasoning.

We conducted extensive performance evaluation experiments to measure the agility as well as the trust level prediction of our trust model. Results indicate that our trust model is capturing and correctly predicting the behavior of nodes at an early stage of the simulation time (i.e., within the first 2% of the number of transactions). Results also show that within the same FCD, our proposed trust model isolates untrustworthy behavior and prevents untrustworthy nodes from affecting the reputation of trustworthy nodes.

As for future directions, we are working on integrating incentive mechanisms to discourage or even prevent untrustworthy behavior. Since honest recommenders are important and vital component in any trust model, incentive mechanisms can be used to encourage honesty and reward honest recommenders.

**Author Contributions:** Conceptualization, M.G. and F.A.; methodology, M.G. and F.A.; software, M.G.; validation, M.G. and F.A.; formal analysis, M.G. and F.A.; investigation, M.G. and F.A.; resources, M.G. and F.A.; data curation, M.G. and F.A.; writing—original draft preparation, M.G. and F.A.; writing—review and editing, M.G. and F.A.; visualization, M.G. and F.A.; supervision, F.A.; project administration, F.A.; funding acquisition, F.A. All authors have read and agreed to the published version of the manuscript.

**Funding:** This project is funded by King Abdulaziz City for Science and Technology (KACST) under the National Science, Technology, and Innovation Plan (project number 13-INF2452-04).

**Institutional Review Board Statement:** Not applicable.

**Informed Consent Statement:** Not applicable.

**Data Availability Statement:** Data sharing is not applicable to this article since no datasets were generated or analysed during the current study.

**Acknowledgments:** The authors would like to acknowledge the support provided by the Interdisciplinary Research Center for Intelligent Secure System and the Deanship of Scientific Research at King Fahd University of Petroleum & Minerals (KFUPM).

**Conflicts of Interest:** The authors declare that they have no conflict of interest.

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
