# Peer review of "Trust-Aware Fog-Based IoT Environments: Artificial Reasoning Approach"

_applsci, doi:10.3390/app13063665_

Round 1

Reviewer 1 Report

The work is goof but the work has some basic things missing in this work. the work seams more mathematical model of trust which does not fits FOG architecture.

1. What is an malicious object what are various features of such node

2. The trust evaluater seems the central code in FOG who will compute it?

 3. The trust is not about malicious object detection but also reducing faults in the network due to these object. More input on this is required

4. The work seems like any existing model of wireless network coated for FOG.

5. How can you defrentiate the working of suck trust model in wireless and FOG.

6. Sharing of trust model will increase the load on network study of this load is required

7.  Missing features to define trustworthy & untrust worthy with their features like fault or security threat? 

8. Trust in FOG is not about simple the detection but based on what behaviour, because a node may be malfunctioning will it be defined malicious?

Author Response

First of all, we would like to thank you for your efforts and cooperation. We would like also to thank the reviewers for their time in reading our manuscript and providing valuable comments to improve the quality of our manuscript.  Here, we are writing this to inform you on how we have addressed all the issues raised by the reviewers. These changes have been incorporated in the revised manuscript.

Reviewer 2 Report

Generally, the paper is well written. Research topic is very timely and interesting. The following are my comments for further improvement of this paper.

1.       The abstract is well presented; however, it does not include the research gap of the existing trust evaluation mechanisms. Some of the interesting findings from the authors’ work in this paper can be also included in the abstract.

2.       The major problem of this paper is to have lack of cohesion. When I look at the title of the paper and abstract, it seems that this paper addresses trust issue which can be applied in any application domain including sensors, fog, or cloud computing. Then when I look into other sections, including related work, this paper mainly considers trust evaluation in fog computing area. Therefore, I suggest revising the abstract and title of the paper so that audience can understand that the paper has research focus on fog computing trust evaluation.

3.     Is the section 2 “Trust Model Architecture Components” part of authors’ contribution? If that is so, please improve the title of this section so that readers can easily understand that it is part of the authors’ contribution in the paper.

4.     The proposed trust model uses subjective logic. Please explain the purpose of using subjective logic instead of any other trust model.

5.     How does a FCD elect a coordinator? What are the criteria for this?

6.     Figure 2 is not clear enough. In this trust reputation operation cycle please clearly explain the steps.

7.     In Section 3, the authors have identified the trust attributes. Is there any motivation for selecting this set of trust attributes?

8.     Please briefly explain the algorithm 1.

9.     Explain in more details how you implemented trust model in contiki OS and evaluated performance in in Cooja simulator.

10.  When there is no direct trust and the recommenders are dishonest how would it affect the performance fog computing platform in terms of services (utilization, blockage, meeting service completion deadline).

11.  Please clarify, in your simulation, whether trust builds up as time goes by. If so, please explain it with relevant results.

12.  Some of the captions in table and figures end with a full stop where some of the captions there is no.  It is important to maintain a consistency throughout the paper.

Typos:

varied the value of λ and the percentages of percentages of malicious

sectionIntroduction The Internet of Things (IoT) is

Author Response

(The authors gave the same response as above.)

Round 2

Reviewer 1 Report

the work has been improved and well presented.